# 30 Days Mortality Prognostic Value of POCT Bio-Adrenomedullin and Proenkephalin in Patients with Sepsis in the Emergency Department

**DOI:** 10.3390/medicina58121786

**Published:** 2022-12-04

**Authors:** Silvia Casalboni, Gabriele Valli, Ferdinando Terlizzi, Marina Mastracchi, Giacomo Fidelio, Francesca De Marco, Caterina Bernardi, Anastasia Chieruzzi, Alessia Curcio, Francesco De Cicco, Nicola Colella, Ilaria Dafne Papasidero, Emanuele Tartarone, Maria Pia Ruggieri, Salvatore Di Somma

**Affiliations:** 1Postgraduate School of Emergency Medicine, Sapienza University of Rome, 00185 Rome, Italy; 2Department of Emergency Medicine, San Giovanni Addolorata Hospital, 00184 Rome, Italy; 3GREAT Network (Global Research on Acute Condition Team), 00100 Rome, Italy; 4Department of Medical -Surgery Sciences and Translational Medicine, Sapienza University of Rome, 00184 Rome, Italy

**Keywords:** sepsis, Bio-ADM, PenKid, POCT IB10, ED, mortality

## Abstract

*Background and Objective:* Sepsis is a worldwide severe disease with a high incidence and mortality rate. Sepsis is a frequent cause of admission to the emergency department (ED). Although prognostic scores (Sequential Organ Failure Assessment, SOFA; New Early Warning Score, NEWS; Rapid Emergency Medicine Score, REMS) are commonly used for risk stratification in septic patients, many of these scores are of poor utility in the ED. In this setting, biomarkers are promising alternatives, easier to perform and potentially more specific. Bio-adrenomedullin (Bio-ADM) and Proenkephalin (PenKid) seem to have a key role in the development of organ dysfunctions induced by sepsis and, therefore, could help in the risk stratification of patients with sepsis at ED admission. The aim of this study was to evaluate the utility of Bio-ADM and PenKid, obtained through a point of care (POCT) device, in predicting 30 days mortality for patients presenting to the ED with sepsis. *Methods and Results:* In total, 177 consecutive adult patients with a diagnosis of sepsis presenting to the ED of San Giovanni Addolorata Hospital in Rome, Italy, between May 2021 and April 2022 were enrolled in this prospective observational study. For each patient, Bio-ADM and PenKid were obtained at ED admission together with SOFA, NEWS and REMS scores. Next, 30 days follow-up data were collected to evaluate patient mortality. Both biomarkers (Bio-ADM and PenKid) and clinical scores (SOFA, NEWS and REMS) were good predictors of mortality at 30 days, with Bio-ADM and REMS outperforming the others. Moreover, PenKid resulted in being linked with the worsening of kidney function. *Conclusions:* In patients presenting with sepsis in the ED, Bio-ADM and PenKid, evaluated with a POCT device, predicted 30-day mortality. These two biomarkers seem even more useful when integrated with clinical risk scores at ED admission.

## 1. Introduction

The incidence of sepsis is increasing worldwide, and it is also linked with high mortality [1], costs [2] and rehospitalization rate, particularly when coupled with septic shock [3].

According to the consensus Sepsis-3, sepsis is defined as life-threatening organ dysfunction caused by a dysregulated response to infection [4]. Consequently, it is crucial to immediately recognize sepsis since a fast diagnosis and treatment represent the best tool to decrease mortality [5]. For these reasons, emergency department (ED) physicians play a central role in the diagnosis, risk stratification and appropriate treatment of patients presenting with sepsis [3,6]. Survival Sepsis Campaign guidelines (SSC) tried to improve the diagnosis and prognosis of sepsis patients, introducing the sequential organ failure assessment (SOFA) and its quick version (qSOFA) [7]. SOFA score has been validated for intensive care unit (ICU) patients, while its usage in the ED presents uncertain results. On the other hand, qSOFA, a score based on only three clinical parameters, is easier and faster to perform in the ED with the drawback of being much less specific and informative [8,9]. In the last few years, the performances of the national early warning score (NEWS) and rapid emergency medicine score (REMS) have been compared to the previously mentioned SOFA and qSOFA in terms of specificity, sensitivity and prediction of adverse outcomes. Performances have been found to be roughly balanced in terms of specificity and sensitivity, with NEWS and REMS scores outperforming SOFA and qSOFA in the prediction of adverse outcomes at 7 and 30 days [8,10]. Despite attempts to develop more accurate scores, none have been shown to be sufficiently reliable for the early recognition of sepsis and its prognosis [5]. Litell et al. evaluated the risk of misdiagnosing sepsis by means of a positive SOFA evaluation, thus administering sepsis therapy and fluid challenge to a non-septic patient, showing how the usage of SOFA alone can even be harmful [11]. Considering this and other similar studies, in 2021, SSC stepped back regarding its SOFA score role in sepsis diagnosis and management, suggesting that clinical evaluation, SOFA, qSOFA and other scores should all be considered [5].

In the last few years, many biomarkers with potential diagnostic or short-term prognostic value for septic patients have been analyzed. Among these, Bio-adrenomedullin (Bio-ADM) and proenkephalin (PenKid) have been shown to correlate with poor outcomes and renal impairment in critically ill patients in the ICU and ED [12,13,14,15,16].

Two recent studies analyzed and validated two assays for Bio-ADM and PenKid processed with a POCT device [17,18]. This technology now allows the physician to perform this analysis in as little as 20 min at the patient’s bedside.

To our knowledge, the prognostic role of Bio-ADM and PenKid, obtained through a POCT system in patients with sepsis in the ED, has not been evaluated yet.

The aim of this study was to evaluate the ability of these two biomarkers, measured via a POCT at the time of the patient’s arrival in the ED, to predict mortality at 30 days and to compare them with classical scores (e.g., SOFA, qSOFA) and biomarkers of sepsis, e.g., procalcitonin (PCT) and C-reactive protein (CRP).

## 2. Materials and Methods

This was a prospective, observational, single-center study that took place in San Giovanni Addolorata Hospital in Rome, Italy. Adult (>18 years) patients with sepsis at the ED between May 2021 to April 2022 were considered eligible. This study was approved by the Ethical Committee of competence (Lazio 2 Ethical Committee, protocol number 0190748, 25 October 2019) and conducted in agreement with the Declaration of Helsinki and its successive amendments. Informed consent was obtained from each patient before enrollment.

### 2.1. Population

Suspicion of severe infection was defined by the presence of the following clinical criteria [19] and risk factors [20,21]. The presence of two clinical criteria or one clinical criterion and at least one risk factor were considered sufficient for enrollment into the study. The diagnosis of sepsis was later confirmed using the current definition of sepsis [5]. The only exclusion criterion considered was the unavailability of consent to the study.
**Clinical Criteria****Risk Factor**fever or hypothermia (considered as body temperature > 38 °C or <35 °C)worsening of mental statuspresence of dyspnea or persistent coughnausea, diarrhea, vomiting or abdominal painurinary tract symptoms, frequent diuresis, dysuria or stranguryleukocytosis or leukopenia (WBC > 12,000 or <4000)intravesical permanent catheter, endovascular or other implantsfrail patientnursing home residentrecent hospitalizationdiabetes mellitusrecent surgical treatment or invasive maneuversimmunosuppression**Inclusion Criteria:** 2 Clinical Criteria or 1 Clinical Criteria + 1 Risk Factor

### 2.2. Study Design

At the time (T_0_) of arrival in the ED, for each patient, after a signed informed consent was obtained, personal data and medical history, clinical evaluation, vital parameters, blood gas analysis (BGA), routine blood tests and blood cultures were collected. Furthermore, blood samples were collected and quickly tested with a POCT device in order to evaluate Bio-ADM and PenKid levels.

After 24 h (T_1_), vital signs were recorded, and routine blood tests, blood cultures and BGA were performed.

PenKid and Bio-ADM levels were obtained with two different Nexus IB10 POCT analyzers, described in detail previously [22]. For every patient, blood was drawn by a peripheral vein line and collected into a single EDTA tube. Within 20 min, using a precision pipette fixed to 500 µL, a patient sample was collected from the EDTA tube and inserted in each Nexus IB10 POCT analyzer. Results were obtained in less than 40 min. The results of the two biomarkers were not utilized for patient management.

Routine laboratory blood tests included complete blood count, bilirubin, creatinine, CRP and PCT and coagulation test, brain natriuretic peptide (BNP), and high sensitivity Troponin I (HsTnI). In addition, clinical, anamnestic and personal information including age, sex, recent hospitalization, recent use of antibiotic therapy, systolic blood pressure (SBP), diastolic blood pressure (DBP), medium arterial pressure (MAP), heart rate (HR), respiratory rate (RR), body temperature (BT), urine output, need for oxygen therapy and if so FiO_2_, need for mechanical ventilation, need for renal replacement therapy (RRT), need for vasopressor and if so, the dosage of catecholamines, fluids infusion in the first 3 h and fluid output were collected for each patient in order to analyze the previously mentioned scores.

Thirty days (T_30_) later, a follow-up phone call was performed to reach patients or family members. Every patient/family member was asked to answer questions regarding the current clinical condition, the day of discharge from the hospital, the eventual occurrence of death or other hospitalizations and when and why they occurred.

### 2.3. Biomarkers Analysis

Biomarkers analysis was performed by the IB10 Sphingotec^®^, (Henningsdorf, Germany) Bio-ADM^®^ [17] and PenKid^®^ assays [18]. These diagnostic assays are built on a disc processed by the Nexus IB-10 analyzer that can be used with ethylenediaminetetraacetic acid (EDTA) whole blood or EDTA plasma specimens, delivering results within 20 min. After the addition of the blood specimen to the test disc, the Nexus IB-10 analyzer provides control of the temperature of the disc, as well as the sequence of centrifugal flow, mixing, incubation time, final signal measurement, quantitation and reporting of results. The centripetal force generated by the rotation of the disc rapidly prepares cell-free plasma, and the combination of active flow and capillary action force it through the microfluidic channel to the immunochromatography module to rehydrate, solubilize and mix with freeze-dried immunoconjugates. The immune complex forms a colored band on the chromatography membrane, and the density of the band proportional to the concentration of the biomarker under analysis in the specimen is quantitatively measured optically. The whole process takes 20 min for each biomarker. The test disc includes a positive internal control to ensure that the test has been performed properly.

Calibration is provided to the user through a 2D barcode affixed to the test disc, which contains lot-specific parameters established during the manufacturing process, and it was performed every day at the beginning of the days before starting the analysis.

### 2.4. Statistical Analysis

Patients were divided into two groups (survivor and non-survivor) according to the outcome at 30 days, and all variables and characteristics were compared between these two groups.

Statistical significance was determined with a threshold of 95% (α < 0.05). The whole statistical examination was performed using the statistical software StatPlus (StatPlus Pro v7©, AlaystSoft Inc., Walnut, CA, US). Continuous variables were shown as the median and interquartile range (IQR), except for age, which was expressed as mean ± SD. Differences among medians of the different groups were tested using the Mann–Whitney U test. Categorical variables were summarized in crosstab, expressed as the % of the group and analyzed using a χ^2^ test. When this test provided a significant result, variables were further analyzed with a z-test. Correlations between variables were analyzed using Pearson regression and verified within a univariate linear regression analysis. The coefficients of linear regression were estimated with the method of minimal squared, and the strength of the correlation was expressed by means of the Pearson coefficient (R). Kaplan–Meier survival analysis was performed in order to evaluate the predictive power of Bio-ADM, PenKid and the clinical scores (SOFA, NEWS and REMS). The cut-off of the variables used for the analysis were 65 pg/mL and 100 pmol/L for Bio-ADM and PenKid, respectively, as previously described by Lundberg et al. [16,23] and Marino et al. [13]. For SOFA, NEWS and REMS, we chose cut-offs of 4, 6 and 6. Differences in survival times were verified with a log-rank test, and the hazard ratio (HR) was calculated for each group analyzed. Survival time was described as mean ± SD, while HR was described as mean ± confidence interval (IC). ROC curves were calculated for each survival predictor, and the AUC of each model was calculated; a test-z was performed to highlight the significant differences between the AUC of each variable.

## 3. Results

### 3.1. Participants and Demographics

A total of 177 consecutive patients with a suspicion of sepsis in ED were enrolled in this study; their characteristics are shown in Table 1.

For each patient, SOFA, NEWS and REMS were calculated at arrival in the ED. The interquartile ranges (IQR) of the scores were, respectively, 4 for SOFA, 6 for NEWS and 6 for REMS. According to the phone call follow-up at 30 days, 24.3% of the patients in our group were deceased. The median survival time was 8 (1–17) days after the emergency department visit. We described the diuresis of our patients: 123 (69.5%) had active diuresis (more than 300 mL per day); in 34 (19.2%), diuresis was compromised: 20 (12%) presented oliguria (between 300 and 100 mL per day), 10 (6%) with anuria (less than 100 mL per day) and 4 enrolled patients were treated with chronic hemodialysis for chronic kidney disease (CKD). The most common septic focus was the lung in 78 (45%) cases; 56 (32%) of the patients had a urinary tract infection, and 14 (8%) had an abdominal infection; in 24 (14%) patients, we found other less common sources of infection. In five patients it was not possible to determine the source of infection in the emergency department. In total, 18 (10,2%) patients required vasopressors to sustain MAP. In all of them, norepinephrine was the inotrope of choice. Our patients often presented other comorbidities, and the IQR for the number of comorbidities resulted in being 2. In our group of patients, 104 (59%) did not require oxygen support, 69 (39%) required oxygen (Venturi mask or nasal cannula) and 4 (2%) were treated with noninvasive ventilation (NIV).

### 3.2. Comparison between Survivor and Non-Survivor Groups

According to the outcome at 30 days, we divided our population into survivors (n. 134, 75%) and non-survivors (n. 43, 25%). In these two groups, demographic and anamnestic information, vital parameters, laboratory findings, biomarkers and scores were tested (Table 2).

Patient age was significantly higher in patients with adverse outcomes. In the non-survivors group, compared to survivors, we observed a significant worst Glasgow coma scale (GCS) and lower MAP as well as more frequent diuresis impairment (33% vs. 16%, *p* < 0.001). Plasma creatinine level and plasma sodium concentration were significantly higher in non-survivals compared to survivals (see Table 2). Non-survival showed a lower median pH and La^−^plasma concentration higher than survival (see Table 2). HsTnI and BNP were significantly higher in non-survivals compared to survivals. No differences were found among the other clinical and laboratory variables. Regarding the biomarkers under study, the medians of Bio-ADM and PenKid were both significantly higher in the non-survivals group compared to the survival group. In particular, the median for Bio-ADM was 44 pg/mL (IQR 44 pg/ML–55 pg/mL) in the survivor group and 55 pg/mL (IQR 45 pg/mL–100 pg/mL) in the non-survivor group (*p*-value < 0.001). For PenKid, the median was 99 pmol/L (IQR 72 pmol/L–167 pmol/L) in survivors and 135 pmol/L (IQR 75 pmol/L–457 pmol/L) in non-survivors (*p*-value < 0.05) (Figure 1, Table 2).

The comparison of the scores between the two groups is also depicted in Table 2: as it is possible to see, each score is significantly higher in the non-survival group (*p*-value < 0.001).

### 3.3. Correlations between Demographic, Clinical and Laboratory Findings and POC Biomarkers

We performed Pearson regression analysis between various anamnestic and laboratory findings and biomarkers tested with the point-of-care device (Table 3).

High levels of PenKid were significantly correlated with lower diuresis, higher SOFA score and higher creatinine levels at arrival. Similarly, high levels of PenKid were related to higher BNP, higher serum potassium, lower pH and higher lactates in blood gas analysis. Bio-ADM levels showed a significant correlation with lower MAP, higher SOFA score, a reduction in urinary output, higher serum creatinine, higher PCT, lower pH and higher lactates in blood gas analysis. Bio-ADM was also significantly correlated to death at 30 days, while we could not find the same for PenKid.

### 3.4. Survival Analysis for Bio-ADM, PenKid and Scores

Kaplan–Meier analysis was used to evaluate survival at 30 days for the biomarkers levels and the scores (SOFA, NEWS and REMS) at the patient’s arrival.

If we consider the biomarkers, Bio-ADM performed better than PenKid when predicting death in 30 days. Patients with Bio-ADM over the cut-off (65 pg/mL) had an HR of 2.14 ± 0.36 for the event of death in 30 days, while patients with PenKid over the cut-off had an HR of 1.84 ± 0.31. In total, 36% of the patients with positive Bio-ADM died within 30 days compared to 20% in the group with Bio-ADM under the cutoff (*p*-value < 0.01). Furthermore, 32% of the patients with PenKid over the cut-off died at 30 days, compared to a figure of 18% for the group with a negative PenKid (*p*-value < 0.05). The Kaplan–Meier curves for Bio-ADM and PenKid are presented in Figure 2A,B.

Every score was capable of predicting adverse outcomes at 30 days. Considering REMS, we set the cut-off value at 6. Patients with a REMS score over 6 points had an HR of 8.4 ± 0.34 for the event of death in 30 days. In total, 31% of the patients with a positive REMS score died in 30 days, while only 4% of the patients with REMS under 6 had the same outcome (*p*-value < 0.0004). Patients with positive REMS had a mean survival time of 21.9 ± 0.92 days. In the survival analysis, REMS was the best-performing tool to predict adverse outcomes at 30 days. The Kaplan–Meier curve for the REMS score is represented in Figure 2E.

Even for the NEWS score, a cut-off of 6 was used. In total, 35% of the patients with NEWS > 6 died, with a mean surviving time of 23.1 ± 1.15 days with an HR of 3.40 ± 0.30. Furthermore, 12% of the patients with NEWS < 6 died with a mean surviving time of 27.5 ± 0.78 days with an HR of 0.29 ± 0.30 (*p*-value < 0.0003). The Kaplan–Meier curve for NEWS is depicted in Figure 2D.

Among the scores, the SOFA score was the one that performed the worst. In the case of SOFA, a cut-off of 4 was used. In the group of patients with a SOFA score > 4, 31% died with an HR of 2.65 ± 0.31 and a mean survival time of 27.1 ± 0.73. In the group with a SOFA score < 4, 13% of the patients died (*p*-value < 0.006). The Kaplan–Meier curve for SOFA score is shown in Figure 2C.

### 3.5. ROC Curves

At last, we performed ROC analysis for the above-mentioned scores and biomarkers (Table 4).

We observed that REMS score and Bio-ADM had the best sensitivity and specificity to predict 30 days mortality with an AUC of 0.73. PenKid was the one with the worst AUC, with a value of 0.60. We also performed a comparison among all these tools, and we noticed that Bio-ADM and REMS performed better than the others with statistical significance (*p* < 0.001). When Bio-ADM and REMS were compared to each other, they appeared equivalent, with none outperforming the other.

Then, we compared the ROC curves of the two novel biomarkers (Bio-ADM and PenKid) with the currently used biomarkers (CRP and PCT). The results of our analysis are shown in Figure 3. The AUC of Bio-ADM was the highest (0.695), followed by PenKid (0.585), with PCT and CRP taking the last places with AUC of 0.511 and 0.479, respectively.

## 4. Discussion

In our analysis, both POCT Bio-ADM and PenKid in patients with sepsis were confirmed to be good predictors of survival at 30 days. These same considerations also apply to main clinical scores (SOFA, NEWS and REMS) [9,10].

Many studies demonstrated that the setting of the ED still lacks a proper tool (be it a score or a biomarker) to screen septic patients according to their mortality risk [8,9,11]. Infection biomarkers such as the CRP and procalcitonin PCT were evaluated in septic patients, but they both seemed not to be specific or sensitive enough for global clinical evaluation. While the levels of these biomarkers can be useful in monitoring and de-escalating antibiotic therapy, the latest guidelines have established that they should not be considered when deciding whether to start antibiotic therapy [5]. Moreover, since the sensitivity and specificity of these biomarkers are not particularly high, in some cases, their usage may induce confusion, and the physician should carefully consider the whole clinical context in the evaluation of these biomarkers [24]. These considerations are even more important because the ED setting is the one that is most frequently involved in sepsis recognition and patients’ risk assessment [3,6]. In this context, our results suggest that Bio-ADM and PenKid, when combined with clinical scores, can represent a potentially easy, quick and informative tool in the identification and risk stratification of septic patients in the ED, particularly when obtained using a POCT system.

Bio-ADM represents the active part of adrenomedullin (ADM), a hormone that is physiologically produced in many cell types and has renal, immunological, endocrine and cardiovascular effects [25,26]. The most interesting effect of ADM is its action on vessels determining both the vasodilation [27] and stabilization of the endothelial barrier acting on endothelial cells to maintain permeability [28,29]. In a 2014 study, Marino et al. showed how plasmatic ADM can predict septic shock and mortality at 28 days in ED septic patients [12]. Recently, some studies have focused on rising levels of Bio-ADM in septic patients, most of which were derived in ICU settings. A recent study analyzed the capability of Bio-ADM to predict adverse outcomes in septic patients, showing how high levels of this biomarker at arrival in the ED are linked to multiorgan failure (MOF) and ICU admission [23]. Findings from our study confirm the result of previous studies [12,15,16], but the present study is the first one to be performed in an ED setting and in which biomarkers were tested with a point-of-care device able to yield results in 20 min. The interest in this biomarker is well justified by the therapeutic possibilities that are offered by the development of a monoclonal antibody, Adrecizumab, able to inhibit Bio-ADM intravascular activity (namely vasodilation) without affecting its endothelial action (membrane stabilizer) [28,30,31]. The clinical implications of the usage of Adrecizumab in humans and septic patients are still to be defined [31]. Using the results obtained with the POCT device, we found higher levels of Bio-ADM in the non-survivor group (Figure 1) and a positive correlation between death and increased levels of Bio-ADM (Figure 2A). The predictive value of POCT Bio-ADM is even more important if we take into account the recent development of the antibody Adrecizumab, which could represent the first molecular physiopathology-addressed tool available in septic patients.

PenKid has been intensively studied as a new biomarker of renal function [32,33] thanks to its special characteristics: it is not bound to plasma proteins, it is exclusively filtered into the glomerulus (neither reabsorbed nor secreted) and is not affected by inflammation or other concomitant diseases [14,34]. For these reasons, PenKid has been proven to be a suitable and accurate marker to estimate the actual glomerular filtration rate (GFR) [35], an early marker of acute kidney injury (AKI) [33] and a prognostic marker in different clinical contexts [13]. Elevated plasma PenKid values predict deterioration of renal function and adverse clinical outcomes in various critically ill patient populations (sepsis and septic shock, heart failure and burns) [13]. As shown in Table 3, in our study, the role of PenKid in mirroring renal function was confirmed by its strong relationship with low urinary output and high levels of creatinine, as well as higher levels of lactates and lower pH. All these parameters have been shown in previous studies to be factors contributing to death and other adverse outcomes (ICU recovery, renal replacement therapy and disability) in septic patients [36]. PenKid levels have also been shown, in previous studies, to rise in blood sooner than creatinine, predicting kidney injury while serum creatinine is still normal [37]. The strong correlation with creatinine has even more importance when considering the possibility of obtaining concrete information on kidney function in septic patients at ED arrival, both for risk assessment and therapeutic decision, with a point-of-care device. It is also fundamental to underline the good performances of the biomarkers in the survival analysis, considering that the most common biomarkers for sepsis evaluation (CRP, PCT) failed to be good predictors of adverse events in our study (Figure 3) [38].

It is challenging to select the right score for sepsis evaluation in the ED. In the present study, we tested three scores to see how they performed when calculated at ED arrival. All of them (SOFA, NEWS and REMS) were significantly increased in the non-survivors group (Table 2) and were good predictors in the survival analysis (Figure 2C–E). The scores confirmed their value in predicting death at 30 days. Among the scores, REMS was the only one initially developed and conceived for ED patients [39]. REMS, similarly to NEWS, does not require blood exams to be performed, so both can be obtained much faster than SOFA scores and are easier to perform in an ED setting. However, it should be considered that a SOFA score above 4 indicates failure of at least one or two organs and that a REMS or NEWS score above 6 indicates a serious impairment of at least two vital functions. On the other hand, rising levels of Bio-ADM and PenKid could be detected much earlier when any impairment of vital function or organ failure is observed.

### Strengths and Limitations

As for our knowledge, this is the first study in sepsis patients using Bio-ADM and PenKid obtained with a POCT settled in the ED. Although this study seems to offer valid conclusions, the results may be limited by the small number of participants (177) and need to be confirmed in the future by larger multicentric studies in the ED.

## 5. Conclusions

This study evaluated the prognostic performance of POCT Bio-ADM and PenKid, in patients with suspicion of sepsis at ED arrival. The results show that, even if both biomarkers performed quite well as predictors of mortality after 30 days, Bio-ADM performed better than PenKid and equally to REMS score. POCT PenKid strongly relates with AKI and creatinine, relevant information of the patient’s kidney function with therapeutic and prognostic potential influence. All the predictive commonly used scores are also to be able to predict mortality, with REMS being the most effective.

Considering our results on Bio-ADM and REMS, it would be important in the future to plan a new study in patients with sepsis in the ED, focusing (e.g., by performing a multivariate analysis) on the additive value of the combination of these two variables.

## Figures and Tables

**Figure 1 medicina-58-01786-f001:**
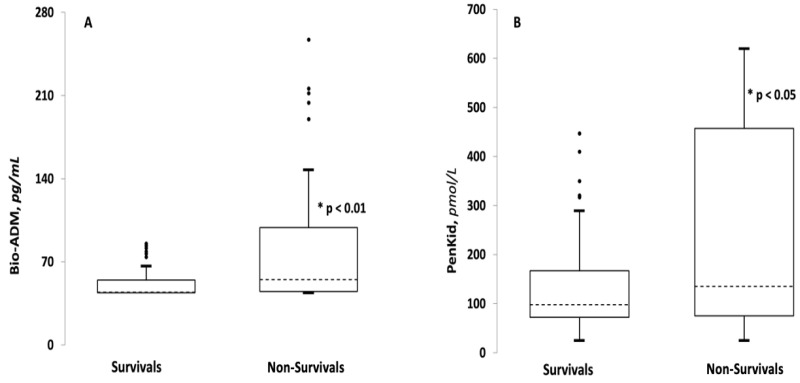
Bio-ADM (**A**) and PenKid (**B**) values in survivals and non-survivals. Median (dash line) and IQR (box) ± 95%CI. Extreme values in bold circles (The dots above the box plots represent outliers values; The * identifies the *p* value).

**Figure 2 medicina-58-01786-f002:**
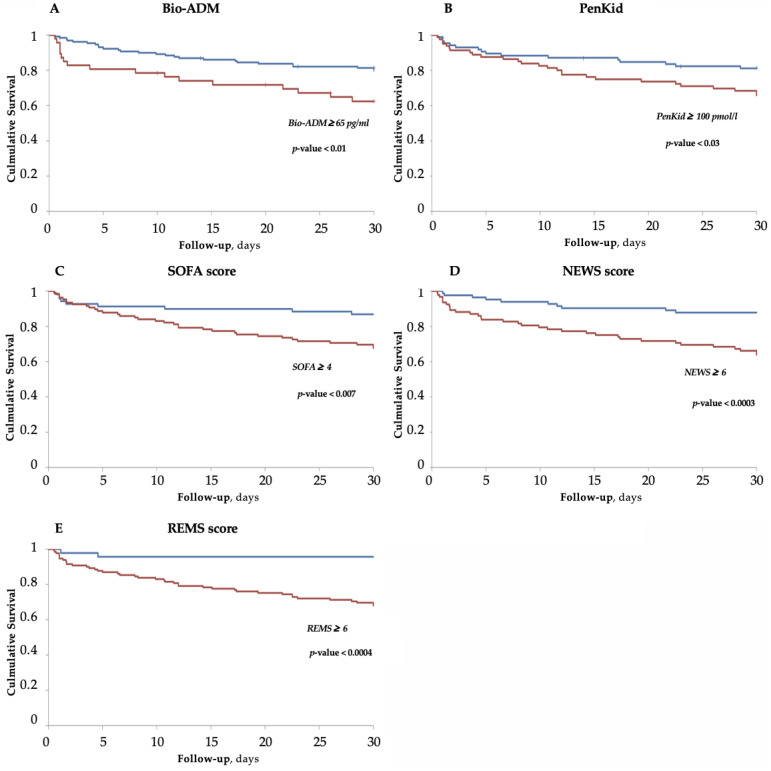
Kaplan–Meier curves according to survival at 30 days for Bio-ADM (**A**), PenKid (**B**), SOFA(**C**), NEWS (**D**) and REMS score (**E**). (**A**) For Bio-ADM, we considered a cut-off of 65 pg/mL. The red curve represents patients with Bio-ADM > 65 pg/mL, the blue curve patients with Bio-ADM < 65 pg/mL; (**B**) for PenKid, we considered a cut-off of 100 pmol/L. The red curve represents patients with PenKid > 100 pmol/L, the blue curve patients with PenKid < 100 pmol/L; (**C**) for SOFA score, we considered a cut-off of 4: the red curve represents patients with SOFA > 4, and the blue curve patients with SOFA < 4; (**D**) for NEWS score we used a cut-off of 6, the blue curve represents patients with NEWS < 6. (**E**) For REMS score we considered a cut-off of 6 points. The dashed curve represents patients with REMS > 6, and the solid curve patients with REMS < 6.

**Figure 3 medicina-58-01786-f003:**
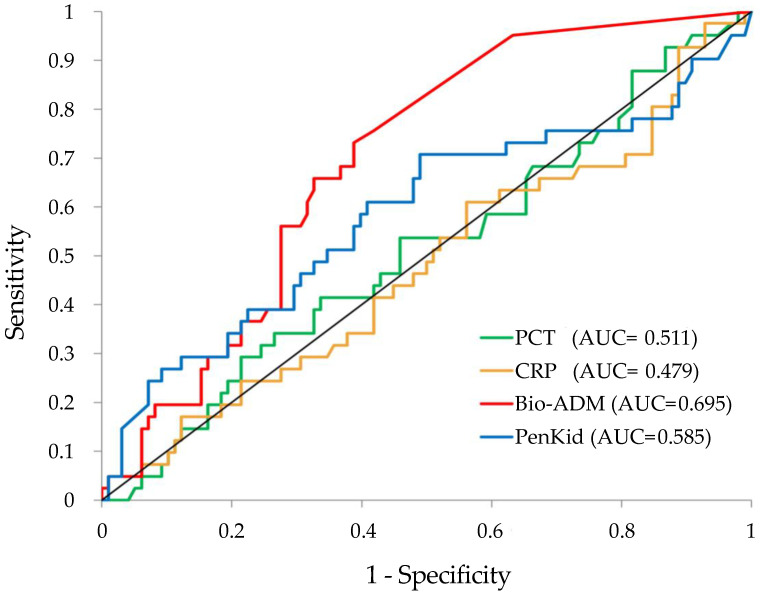
ROC curves for biomarkers. Bio-ADM: Bio-adrenomedullin, red line; PenKid: proenkephalin A, blue line; PCT: procalcitonin, green line; CRP: C reactive protein, yellow line.

**Table 1 medicina-58-01786-t001:** Patients characteristics.

Patients		Characteristics
n.		177
Age, years ± SD		73.1 ± 17.3
Female, n (%)		98 (55.4%)
SOFA, points (IQR)		4 (2–6)
NEWS, points (IQR)		6 (3–9)
REMS, points (IQR)		6 (5–9)
Deaths, n (%)		43 (24,3%)
Comorbidity, n (IQR)		2 (1–3)
Follow-up in Non-Survival, days (IQR)		8 (2–17)
Diuresis:	Active, n (%)	123 (69.5%)
	Oliguria, n (%)	20 (12%)
	Anuria, n (%)	10 (6%)
	Dialysis, n (%)	4 (2%)
Vasopressor, n (%)		18 (10.2%)
Site of infection:	Lung, n (%)	78 (45%)
	Urinary Tract, n (%)	56 (32%)
	Abdominal, n (%)	14 (8%)
	Others, n (%)	24 (14%)
O_2_ support:	NIV, n (%)	4 (2.4%)
	VMK, n (%)	70 (42.7%)
	AA, n (%)	90 (54.9%)

SOFA, Sequential Organ Failure Assessment; NEWS, National Early Warning Score; REMS, Rapid Emergency Medicine Score; NIV, Non-Invasive Ventilation; VMK, Venturi Mask; AA, ambient air.

**Table 2 medicina-58-01786-t002:** Comparison between survivor and non-survivor groups.

	Survivors	Non-Survivors	*p*-Value
N	134	43	
Age, years (IQR)	74 (59–83)	85 (79–90)	<0.001
Female, n (%)	60 (45%)	19 (44%)	n.s.
Comorbidity, n (IQR)	2 (1–3)	2 (1–3)	n.s.
Active Diuresis, n (%)	113 (84%)	29 (67%)	0.01
GCS, point (IQR)	15 (14–15)	14 (12–15)	<0.001
RR, breath/min (IQR)	20 (18–22)	21 (18–27)	n.s.
HR, bpm (IQR)	94 (82–105)	90 (84–100)	n.s.
MAP, mmHg (IQR)	88 (78–100)	77 (69–99)	0.03
BT, °C (IQR)	37 (36–38)	36 (36–38)	n.s.
SpO_2_, % (IQR)	96 (94–98)	96 (92–97)	n.s.
P/F mmHg/% (IQR)	350 (264–405)	291 (214–396)	n.s.
pH, n (IQR)	7.44 (7.40–7.48)	7.42 (7.30–7.46)	0.02
La^−^, mmol/L (IQR)	1.8 (1.4–2.6)	2.2 (1.8–3.8)	0.004
Creatinine, mg/dL (IQR)	1.0 (0.8–1.5)	1.6 (1.1–2.5)	<0.001
K^+^, mmol/L (IQR)	3.9 (3.5–4.3)	4.1 (3.5–5.0)	n.s.
Na^+^, mmol/L (IQR)	136 (133–139)	139 (135–144)	0.003
CRP, µg/dL (IQR)	14 (5–25)	14 (3–25)	n.s.
Glucose, mg/dL (IQR)	122 (101–156)	129 (101–179)	n.s.
BNP, pg/mL (IQR)	162 (36–336)	369 (254–483)	<0.001
HsTnI, ng/L (IQR)	15 (5–44)	55 (26–138)	<0.001
PCT, ng/mL (IQR)	0.9 (0.2–7.0)	1.3 (0.3–14.0)	n.s.
SOFA, point (IQR)	4 (2–6)	6 (4.5–7)	<0.001
NEWS, point (IQR)	5 (2–8)	8 (6–10.5)	<0.001
REMS, point (IQR)	6 (5–8)	8 (7–10)	<0.001
Bio-ADM, pg/mL (IQR)	44.5 (44–54.575)	55 (45.05–98.75)	<0.001
PenKid, pmol/L (IQR)	97.85 (72.1–167)	135.2 (75.3–457.4)	0.05

GCS, Glasgow coma scale; RR, respiratory rate; HR, heart rate; MAP, medium arterial pressure; BT, body temperature; SpO_2_, peripheral saturation of oxygen; La-, lactates; CRP, C reactive protein; BNP, brain natriuretic peptide; HsTnI, high-sensitivity Troponin I; PCT, procalcitonin; SOFA, sequential organ failure assessment; NEWS, national early warning score; REMS, rapid emergency medicine score; Bio-ADM, Bio-adrenomedullin; PenKid, proenkephalin A.

**Table 3 medicina-58-01786-t003:** Pearson analysis for PenKid and Bio-ADM.

	PenKid		Bio-ADM	
	*R*	*p*	*R*	*p*
Age, years	0.18	0.01	0.13	n.s.
Comorbidity, n	0.18	0.02	0.2	0.007
MAP, mmHg	−0.13	n.s.	−0.27	<0.001
Compromised diuresis	0.35	<0.001	0.32	<0.001
SOFA	0.27	<0.001	0.2	0.006
pH, n	−0.27	<0.001	−0.31	<0.001
La^−^, mmol/L	0.35	<0.001	0.47	0.001
K^+^, mmol/L	0.25	<0.001	0.21	n.s.
Creatinine, mg/dL	0.53	<0.001	0.26	<0.001
BNP, pg/mL	0.32	0.002	0.12	n.s.
Hs-TnI, ng/L	0.01	n.s.	0.01	n.s.
PCT, ng/mL	0.16	n.s.	0.21	0.01
Death	0.15	0.05	0.35	<0.001

MAP, medium arterial pressure; SOFA, sequential organ failure assessment; La^−^, lactates; K^+^, potassium; BNP, brain natriuretic peptide; Hs-TnI, high-sensitivity Troponin I; PCT, procalcitonin; PenKid, proenkephalin A; Bio-ADM, Bio-adrenomedullin.

**Table 4 medicina-58-01786-t004:** ROC curves for scores and biomarkers.

	AUC	SE	LCL (98%)	UCL (98%)
SOFA	0.68 ^#, $, °^	0.05	0.57	0.79
NEWS	0.68 ^#, $, °^	0.04	0.57	0.79
REMS	0.73 *^, §, °^	0.04	0.63	0.82
PenKid	0.60 *^, §, #, $^	0.06	0.46	0.73
Bio-ADM	0.73 *^, §, °^	0.04	0.63	0.82

* *p* < 0.001 vs. SOFA; ^§^ *p* < 0.001 vs. NEWS, ^#^
*p* < 0.001 vs. REMS, ° *p* < 0.001 vs. PenKid, ^$^
*p* < 0.001 vs. Bio-ADM. SOFA, sequential organ failure assessment; NEWS, national early warning score; REMS, rapid emergency medicine score; PenKid, proenkephalin A; Bio-ADM, Bio-adrenomedullin.

## Data Availability

The data presented in this study are available on request from the corresponding author. The data are not publicly available due to patient confidentiality.

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
