# Peer review of "30 Days Mortality Prognostic Value of POCT Bio-Adrenomedullin and Proenkephalin in Patients with Sepsis in the Emergency Department"

_medicina, 2022, doi:10.3390/medicina58121786_

Round 1

Reviewer 1 Report

In Figure 1, it is necessary to make the signatures more legible (larger)

Reviewer 2 Report

This prospective observational study on the prognostic value of POCT Bio-ADM and proenkephalin in patients with sepsis in the ED, attempts risk stratification of sepsis patients with a POC tool. It is quite interesting and may be useful in the clinical setting of the ED. The manuscript is well-written. The background is clearly given in a concise and informative Introduction.  The aim is explicitly stated. The methods are clear and the results are comprehensible. The Discussion is well written and adequately covers the topic. However, there are some parts that may be improved by revision. Please see my comments below:

1. Material and Methods:

a) Did your study protocol conform to the Declaration of Helsinki and its successive amendments? If so, you should add this information here.

b) According to the current definition for sepsis (SEPSIS-3, Singer et al, 2016), patients with suspected infection and organ dysfunction (i.e. qSOFA ≥2, or SOFA ≥2) have sepsis. Did you apply SEPSIS-3 criteria to screen your study population? In that case you should tag the table with clinical criteria and risk factors as “Table 1: Criteria and risk factors for suspected infection” removing the criterion of qSOFA and SOFA. Then sepsis is diagnosed in those patients with suspected infection who have a qSOFA ≥2, or SOFA ≥2.

However, in the case that you didn’t apply the SEPSIS-3 criteria (probably due to the recommendation against the use of qSOFA as a single screening tool for sepsis in the recent SSC Guidelines by Evans et al, 2021), please add the relative studies that support your choice of diagnostic criteria for sepsis (e.g. 2 clinical criteria of your list, or 1 clinical criterion and 1 risk factor) and explain how you came up with this list  in the text.

2. Results

a) Comparison between survivor and non-survivor groups:

Please add that you have divided your population in survivors and non-survivors, according to the outcome at 30 days from the inclusion to the study.

b) Figure 1. Please use larger text in the x and y axes of the graphs

c) Table 2: Heading: Correct “comparation” with “comparison”. You should also add the values of Bio-ADM and PenKid along with the p-values in this Table. You may also consider reporting Risk scores in Table 2, and remove Table 3. In that case you need to renumber all your Tables.

d) Table 4. You probably mean “Compromised diuresis” instead of “Diuresis”.

e) Page 8. You should note that most correlations in Table 4 are significant due to p values <0.05. However, the strength of a significant correlation depends on the value of the Pearson correlation coefficient r: r>0.6, strong correlation; r=0.4-0.6, moderate; r<0.4, weak correlation. All correlations in Table 4 are weak, except creatinine to PenKid which is moderate. Therefore you should correct the following text:

 “High levels of PenKid resulted to be strictly related with… “. You should correct this phrase: “High levels of PenKid were significantly correlated with…..

“BioADM levels showed a strong correlation with lower SBP….” Please correct this phrase to “BioADM levels showed a significant correlation with lower MAP (not SBP)…”.

“Bio-ADM is also strongly related to death at 30 days…” Correct this as follows: “Bio-ADM is also significantly correlated to death at 30 days…”

f) 3.4. Survival analysis for Bio-ADM, PenKid and risk scores.

Kaplan-Meier analysis does not evaluate the “effect” of biomarkers on survival, but rather estimates the survival over time in association with the biomarkers or risk scores. Please correct the phrase. Also add (Figure 2), but remove the scores SOFA and NEWS from the sentence. In Figure 2 you only depict Kaplan-Meier estimates for REMS.

In order to evaluate the significance of Kaplan-Meier curves, you should also report the log-rank test with p-value. Please add this in the Figure legend.

g) 3.5. ROC curves: Here you may consider adding a Figure with comparative ROC curves of various biomarkers. It would be interesting to also include in the ROC analysis the inflammatory biomarkers CRP and procalcitonin, to compare them with Bio-ADM and PenKid.

3. Discussion

a) Page 10: Please correct the phrase “ These considerations are made even worse… with : “These considerations are even more important …”

b) Page 11: You state: “It is also fundamental to underline …. (CRP, PCT) both failed in our study to be good predictors of adverse events”. In order to support this conclusion, you should have performed a ROC analysis and a survival analysis for CRP and PCT. I strongly suggest that you do.

c) You state: “All of them (SOFA, NEWS and REMS) are significantly increased …. and are good predictors in the survival analysis”. To support this statement, you should first add the Kaplan-Meier survival analysis for SOFA and NEWS in Figure 2 (these 2 are missing) and second you should report the log-rank test in the results. This test explores whether the difference between survival times between two groups is statistically different or not.

d)You state that “The scores confirmed their value in predicting death ….especially when coupled with biomarkers”. Yet, you do not report any analysis for the combination of a risk score with other biomarkers (Bio–ADM or PenKid). You may add this analysis and report your results for the combination of REMS with Bio-ADM.

4. You should add a separate paragraph at the end of your Discussion to clearly state the strengths and limitations of your study. This is an important part of every manuscript!

5. Conclusions

Please highlight your main finding, that Bio-ADM performed better than PenKid and equally to REMS  in predicting 30-day mortality.

You last sentence regarding the limitations of your study should be moved to the end of the Discussion and be explained in  more detail.

6. The English language and grammar needs to be corrected in many instances. For example:

a) Abstract, Methods and results:

The phrase “ In this prospective….. April 2022” needs correction: “177 consecutive adult ……..sepsis were enrolled in this …study”

Replace the phrase “resulted to be” with a verb, such as “was” (e.g. “were good predictors”)

Replace the phrase “resulted to be able to predict” with a verb, such as “predicted 30-day mortality

b) Introduction “As consequence it is crucial to be able”: Replace with “Consequently, it is crucial to immediately recognize sepsis”

c) Table. Clinical criteria: Frequent urination may be a more known term than “pollakiuria”. Risk factors: replace “compromission of immunity system” with “immunosuppression”

d) Study design: Replace “anamnestic information”  with “medical history” throughout the text. Use “vital signs were recorded” instead of “vital parameters were collected”. PenKid and Bio-ADM levels were obtained instead of “dosages”

e) Table 1: “Sepsis focus”: a better term would be “site of infection”

f) Results: “24,3% of the patients in our group resulted to be dead”. It would be more appropriate to say “ were deseased”.

g) Page 6: we divided our population in survivors (instead of between), Patients’ age was significantly…. instead of “resulted”.

h) Page 7: Correct this: The comparison of the scores ….is depicted in Table 3. Correlations between demographic, clinical and laboratory findings….

i) Page 11: “The predictor power” should be “ the predictive value”.

j) “thanks to its peculiar characteristics” should be “special”

k) “As it is possible to see in Table 2” should be “ As it is shown in Table 4”

l) “Predicting kidney damage” should be “kidney injury while serum creatinine is still normal”
